# Biodistribution of a Mucin 4-Selective Monoclonal Antibody: Defining a Potential Therapeutic Agent Against Pancreatic Cancer

**DOI:** 10.3390/ijms26136042

**Published:** 2025-06-24

**Authors:** Achyut Dahal, Jerome Schlomer, Laura Bassel, Serguei Kozlov, Joseph J. Barchi

**Affiliations:** 1Glycoconjugate and NMR Section, Chemical Biology Laboratory, The Center for Cancer Research, National Cancer Institute at Frederick, Frederick, MD 21702, USA; achyut.dahal@nih.gov; 2Center for Advanced Preclinical Research, Frederick National Lab for Cancer Research, The Center for Cancer Research, National Cancer Institute at Frederick, Frederick, MD 21702, USA; jerome.schlomer@nih.gov (J.S.); laura.bassel@nih.gov (L.B.); kozlovse@mail.nih.gov (S.K.)

**Keywords:** mucin, glycosylation, glycopeptide, monoclonal antibody, near-infrared dye, biodistribution, pancreatic ductal adenocarcinoma

## Abstract

We have previously reported on a novel monoclonal antibody (mAb) we designated F5, which was raised against a glycopeptide derived from the tandem repeat (TR) region of Mucin-4 (MUC4), a heavily O-glycosylated protein that is overexpressed in many pancreatic cancer cells. This mAb was highly specific for the MUC4 glycopeptide antigen in glycan microarrays, ELISA and SPR assays, selectively stained tissue derived from advanced-stage tumors, and bound MUC4^+^ tumor cells in flow cytometry assays. The mAb was also unique in that it did not cross-react with other commercial anti-MUC4 mAbs that were raised in a similar but non-glycosylated TR sequence. Here we describe the selective conjugation of a novel near-infrared dye to this mAb and in vivo biodistribution of this labeled mAb to various MUC4-expressing tumors in mice. The labeled mAb were selectively distributed to both cell-derived xenograft (CDX) flank tumors and patient-derived xenograft (PDX) tumors that expressed MUC4 compared to those that were MUC4-negative. Organ distribution analysis showed high uptake in MUC4^+^ relative to MUC4^−^ tumors. These results suggest that mAb F5 may be used to develop MUC4-targeted, passive antibody-based immunotherapies against Pancreatic Ductal Adenocarcinomas (PDACs) which are notorious for being refractory to many chemo- and radiotherapies

## 1. Introduction

Pancreatic cancer (PC) is the third leading cause of cancer deaths worldwide and is becoming more prevalent. About 90% of PCs are characterized as Pancreatic Ductal Adenocarcinomas (PDACs) that originate in the exocrine tissue of the pancreas. PC is one of the most deadly cancers due to several confounding factors [1]: (1) The anatomical positioning of the pancreas makes it a very challenging organ to operate on. (2) PDAC tumors are poorly vascularized and highly fibrotic making drug access difficult. (3) In addition, the tumor microenvironment is highly immunosuppressive. Most importantly (4) the disease can remain asymptomatic until very late stages of the malignant progression, when metastasis has already occurred, rendering any surgical resection impossible. There is thus an obvious need for enhanced methods of early detection and more effective therapeutic options for PC.

First-line treatment for patients presenting with borderline or non-resectable PDAC is chemotherapy using the FOLFIRINOX protocol (5-fluorouracil, oxaliplatin, irinotecan, and leucovorin) followed by gemcitabine and/or radiation as an adjuvant therapy. These therapies do extend survival but suffer from low overall efficacy and powerful adverse effects. As in many other cancers, if PDAC is diagnosed at late stages, the chances of survival are very slim. The overall 5-year survival rate for those diagnosed with PC was around 8–9% a decade ago; this has risen to ~13%, primarily due to improvements in early detection methods and advances in immune-based therapies [2]. Both passive and active immunotherapies hold promise for the treatment of both early- and advanced-stage PDAC, and several new agents are in pre-clinical/clinical development [3,4,5,6,7]. In addition, for these therapies to be effective, highly tumor-specific antigens need to be identified to allow selective killing of PDAC tumor cells while sparing the surrounding normal tissue and minimize off-target toxicity [8].

Mucins are a family of >20 proteins that coat the surface of epithelial cells and play important roles in cellular homeostasis. Mucins are very large and highly O-glycosylated molecules that form gel-like structures (mucus) that protect cells from outside insult and hence are critical for antimicrobial action in many tissues. They are also involved in crucial signaling processes, cellular adhesion, and tissue organization. Mucins contain a long, extended extracellular domain comprised of multiple tandem repeat (TR) peptide units that may repeat hundreds of times. The TRs range from ~16 to 24 amino acids in length and contain multiple serine (Ser), threonine (Thr) and proline residues. The Ser/Thr hydroxyl groups are nearly all O-glycosylated; glycan transfer to these sites is appropriately referred to as mucin-type glycosylation and this particular modification always begins with an α-linked galactosamine unit (α-GalNAc) (see green box, Figure 1). As is the case in all neoplasms, the glycan content of tumor-associated mucins is dramatically modified relative to the normal phenotype, primarily through truncation of these O-linked saccharide chains. These are referred to as Tumor-Associated Carbohydrate Antigens (TACAs) and, since they are specific structures presented only on tumor cell surfaces, they have been targets for immunotherapy against many cancers [9]. Mucins (abbreviated with the letters MUC) that are overexpressed in many cancer types include MUC1, MUC5AC, and MUC16, and many of these are relevant to PDAC progression [10,11,12,13]. A now well-established target in PDACs is Mucin 4 (MUC4), a large cell surface mucin that is exclusively expressed in pancreatic neoplasms but not in normal pancreatic tissue (Figure 1, center) [14,15,16,17]. The MUC4 protein contains several domains, each of which may promote tumor progression. For example, MUC4 Epidermal Growth Factor (EGF) domains interact with EGF Receptor (EGFR) ectodomains to keep mitogenic signaling active [18]. Aberrant glycosylation is also a tumor-associated feature of the MUC4 protein, and certain TACAs are known to be present in the TR region. One of these is the Thomsen Friedenreich antigen [9,19,20,21,22,23,24]. (TF_ag_), also known as the core-1, O-linked, mucin-type glycan. It is a simple disaccharide (viz., Galβ1-3GalNAcα-*O*-Ser/Thr) found almost exclusively on tumors, contributes to metastasis and aggression, and is a target for tumor vaccine and immunotherapy design (Figure 1, green box) [9,25,26,27,28,29,30,31,32,33,34].

A previous study in our lab employed gold nanoparticles (AuNPs) as a vaccine platform to immunize mice with a set of MUC4 TR glycopeptides containing the TF_ag_ at various O-linked positions [35]. The results of this study showed that one glycopeptide (MUC4-Ser^5^-TF_ag_) generated a good immune response, and sera from animals vaccinated with this construct was 100% specific for the antigen and other MUC4-TR peptide analogs on a large glycan microarray. Based on these data, we developed a monoclonal antibody (mAb) to this glycopeptide called F5. This commercially available (Rockland Immunochemicals) mAb was shown to specifically stain PDAC tissue, bind tumor cells expressing aberrant MUC4, and maintain its specificity against a panel of many TF_ag_-containing molecules on much larger glycan microarrays with a much broader range of components [36]. In this paper, we show how F5 labeled with a near-infrared dye can target MUC4-expressing tumors in mice and is selective for MUC4^+^ tumor over those that are MUC4^−^. This behavior extends to patient-derived xenograft (PDX) models that are MUC4-positive. Appropriate purification and characterization of the conjugates is also critical to preventing accumulation in other organs outside of the tumor.

## 2. Results

The work of Batra and others have established MUC4 as a highly specific marker of PDACs, as this protein is rarely (or never) found on normal pancreatic cells. The expression of MUC4 also seems to corelate with pre-malignant disease stages (pancreatic intraepithelial neoplasms, PanIN lesions) and increases as tumor aggressiveness increases [37]. We recently developed a new mAb that binds MUC4^+^ tumor cells based on the results of a vaccination study performed with several “random” glycopeptides derived from the TR region of MUC4. The glycopeptides contained TF_ag_ at various serine and threonine positions. TF_ag_ is a TACA that is found primarily in tumors and has been shown to be present on tumor-associated MUC4 proteins [38]. Thus, in PDACs, as is in many other cancers, the first structures subject to immune surveillance are glycosylated peptide backbones of proteins. It goes to reason that the immunogens in a vaccine construct should contain the glycan in the proper attachment to the peptide backbone. The Batra group has prepared MUC4-targeted antibodies raised against both the TR region [39] and the non-TR sections [40] of the protein. The TR region mAb was raised to a similar peptide sequence as what we used in our work, only frame-shifted and unglycosylated. This binds to both normal and PDAC cells. Our mAb raised to a glycosylated TR construct was virtually tumor-specific. One conclusion that can be drawn from these results is that mimicking the “immunological landscape” of a PDAC tumor cell surface necessitates a hapten that most closely resembles what is exposed on the cell surface. In its simplest form, such haptens would be represented by the primary structures of extracellular protein motifs that include their post translational modifications (i.e., glycosylation).

### 2.1. Labeling and Characterization of F5-FNIR Conjugate

We used the Frederick Near-Infrared dye (FNIR) developed by the Schnermann lab here at NCI Frederick [41]. This is a very bright, non-aggregating dye that has been used in many in vivo applications. The dye conjugation reaction followed standard protocols by mixing F5 mAb with varying concentrations of FNIR-NHS (Figure 1) to achieve different degrees of labeling (DOL). Table 1 lists the calculated data used to determine the DOL for the FNIR conjugates. The labeled mAb was characterized by UV spectroscopy and DOL was calculated based on protein concentration and UV absorbance at both 280 and 765 nm (See Section 4). This was also corroborated by MALDI-TOF Mass Spectrometry (Appendix A). SDS page gel analyses were run to confirm the covalent conjugation of dye with antibody by fluorescence imaging. Conjugates with a DOL of 3–4 were optimum and used for in vivo biodistribution studies. Size Exclusion Chromatography (SEC) was also used to confirm the conjugation of dye to the F5 antibody (Figure 2).

Near-IR dyes are complex molecules that are amphiphilic and contain functional groups that may interact with other molecules in vitro/in vivo or impart modified properties to biomolecules such as mAbs. We undertook binding studies of F5-FNIR to ensure no detrimental effects on the binding of antibody to antigen after conjugation. For this we used both Surface Plasmon Resonance (SPR) and ELISA assays using our optimized immunogen, which had a specialized linker attached to the N-terminal side of the glycopeptide and terminated with a thiol moiety (See Figure 1) [36]. Analysis of the SPR data in Figure 3A revealed an apparent K_d_ of approximately 2.70 nM for the labeled mAb (F5-FNIR) to its antigen. This was comparable with the parent F5 antibody (1.04 nM), confirming that there was essentially no effect on binding after dye conjugation (Figure 3B and Appendix A). The isotype control mAb showed no binding to antigen (Figure 3C). ELISA studies confirmed that F5-FNIR with different DOL’s binds to the antigen almost identically to that of the unlabeled F5 (Figure 3D).

### 2.2. Biodistribution Studies

We sought to determine the ability of F5 to distribute selectively, in vivo, to appropriate tumor tissues, i.e., those that are MUC4^+^. We chose three tumor cell lines: Two with documented expression of MUC4 (HPAF-II and Capan-1) and one that was known not to express MUC4 (PANC-1). Cells were grown under conditions outlined in the Materials and Methods section, and subsequently analyzed for MUC4 expression by Western blots, utilizing F5 as the detection mAb. Those known to be MUC4-positive all stained well, while PANC-1 was confirmed negative for MUC4 expression (Appendix A). All three tumor lines were subsequently injected into the flanks of immuno-compromised mice. All implants displayed vigorous outgrowth in recipient animals and 18 days after implantation, F5-FNIR was administered by tail vein injection. Mice were analyzed for fluorescence and individual organs were examined upon animal necropsy and dissection. Figure 4 shows the in vivo-normalized fluorescence localization over time on MUC4^+^ and MUC4^−^ tumor in mice. F5-FNIR accumulated in the tumor tissues expressing high levels of MUC4, whereas much less accumulation was detected in the MUC4^−^ tumor (Figure 4).

Animals were sacrificed, and organs were examined for dye uptake. Figure 5 shows that the accumulation was primarily in the MUC4^+^ tumor with some liver involvement, which may be due to the hydrophobicity of the conjugated dye molecule. No other organs appeared to contain detectable amounts of labeled mAb.

A similarly dye-labeled standard isotype control mAb (IgG-FNIR) displayed low distribution to tumors regardless of their MUC4 expression status. Appendix A shows the in vivo (Appendix A, top) and ex vivo (Appendix A, bottom) fluorescence study of F5-FNIR and IgG-FNIR in the Capan-1 CDX model, similar to what is shown in Figure 4 and Figure 5. During other unpublished studies with F5, a third MUC4-positive cell line, BxPC-3, was used for further validation of the selective biodistribution of F5-FNIR in BxPC3-derived flank tumors in mice. Similar to the previous study, we observed the localization of F5-FNIR conjugates to these tumors relative to isotype control (Appendix A).

### 2.3. Patient-Derived Xenograft Models

The results of the cell-derived xenograft biodistribution studies prompted us to determine if similar uptake would take place in PDX models of PDAC tumors. To this end, we interrogated the NCI Patient-Derived Models Repository, a database of PDX tissue maintained by the Division of Cancer Treatment and Diagnosis (DCTD) of the NCI (https://pdmr.cancer.gov/database/default.htm (accessed on 5 January 2025)) to identify PDAC PDX models displaying various abundance in MUC4 expression. The PDMR resource contains a wealth of information regarding the tumor origin and patient diagnosis, as well as transcriptome analysis data of the tumor tissue. This allowed us to identify several samples that were highly MUC4^+^ as well as those that did not express MUC4. A MUC4^+^ PDX model #133R and MUC4^−^ model #295R were selected for subsequent work. Tissue immunohistochemistry staining with the F5 mAb confirmed MUC4 expression in #133R tumor tissue and lack of detectable MUC4 in #295R tissue samples. Figure 6 shows both the entire tumor slice and an expanded section of 133R (Figure 6A,B,E,F) and 295R (Figure 6C,D,G,H) where the upper panels are stained with F5 and the lower panels with isotype control mAb. Figure 6A,B shows strong cytoplasmic and membranous staining of the neoplastic cells in the 133R sample and virtually no staining of 295R (Figure 6C,D). Figure 6E–H show very little staining with the isotype control. These results were consistent across multiple tumors evaluated showing statistically significant differences in MUC4 percent positive and H-score (Appendix A).

These were both grown in mice and, and although established tumors were formed with MUC4^+^ PDX 133R, 295R growth kinetics were much slower and less consistent. Thus, mice engrafted with the #295 tissue were not included in the final analysis.

As observed in the experiments with the cell-derived tumors, tail vein injection of F5-FNIR into animals with tumors from #133R PDX confirmed good distribution of F5-FNIR to tumors at a level superior to that of the IgG isotype control (Figure 7). However, in this experiment, animals receiving the FNIR isotype control conjugate revealed a somewhat higher level of fluorescence in xenograft tumor tissues than cell-derived tumors. We surmised that the isotype control mAb may be binding antigen within the tumor and decided to test this hypothesis by immunohistochemical staining using the isotype control antibody. A direct comparison of F5 and isotype control staining on tissues between PDX models #133 and #295 demonstrated that the isotype control IgG exhibited a marginally higher level of background staining in this tissue relative to the F5 antibody (Figure 6B,D). However, the extent of background staining observed in tumor tissues was at a similar level to other tissues, indicating that preferential binding of isotype antibody to tumor tissues was not occurring. The fluorescence signal observed in tumors treated with the isotype control antibody is, therefore, attributable to non-antigen-specific accumulation within the tumor microenvironment. This signal accumulation more likely resulted from a combination of the enhanced permeability and retention (EPR) effect characteristic of tumors and uptake by tumor-associated macrophages.

We observed high distribution to the liver in the cell-derived tumors (*vide supra*). This could be associated with unwanted adverse effects in a therapeutic setting, e.g., through off-target toxicities from F5 antibody-based therapies (*vide infra*). We surmised that if the FNIR conjugates are not properly purified, higher-order aggregates or residual-free dyes may be present, which could lead to problematic effects. Prior to the PDX biodistribution experiments, we extensively purified the dye conjugates by SEC. After sacrificing the animals and conducting necropsies of individual organs, we found that there was reduced liver involvement than in the experiments with the cell-derived tumors (Figure 8). As stated above, working with modified biologic drugs like mAbs necessitates high purity and full characterization for any experiments to be interpreted properly.

## 3. Discussion

Recent advances in immunotherapies against cancer have transformed the anti-tumor therapeutic landscape. Some of these modalities include mAbs, bispecific mAbs, antibody–drug conjugates (ADCs), a variety of different CAR-T cells, small-molecule immunomodulators, and many novel cancer vaccine platforms targeting repertoires of neoantigens [42]. Many of the approved therapeutic interventions are for blood-borne or “liquid” tumors. It has been much more challenging to reach clinically relevant efficacy for solid tumors. Some of the issues include, but are not limited to [43,44], (1) low expression of tumor-specific antigens, (2) stromal barriers to tumor penetration, (3) T-cell exhaustion, (4) down-regulation of antigen and off-target toxicity, and (5) the immunosuppressive tumor microenvironment. There are many new clinical trials of immunotherapeutic platforms for many different solid tumors, and the hope is that some of these new strategies will be clinically validated and approved for use soon. Until then, novel approaches to new therapeutic interventions are desperately needed.

Here we present the murine biodistribution details of a mAb that was raised to a glycosylated sequence of the TR region of MUC4. We were able to show that this mAb homes selectively to tumors derived from established cell lines as well as those derived from patient tumor biopsies. In addition, organs were for the most part devoid of dye-conjugated F5 mAb, except for the liver in our initial studies. This liver deposition was greatly reduced when the conjugated mAb was fully purified to homogeneity before injection into animals. As far as we know, this is the first instance of a mAb raised to a glycosylated TR sequence from MUC4 to be analyzed for tumor-targeting in a live animal system. A very recent study by Jaiswal et al. showed another MUC4-targeting mAb that can distribute to orthotopic models of PDACs in athymic nude mice [45]. The details of the mAb used in this study were not disclosed, although it was most likely one raised to an unglycosylated TR sequence, similar to the sequence depicted in Figure 1, only frame-shifted (i.e., STGDTTPLPVTDTSSV [46]). Also, the two tumor lines used in the Jaiswal study (SW1990 and CD18/HPAF) were different from ones used in our present work (Capan-1, HPAF-II, and BxPC3) and no control MUC4-negative cell line was analyzed. The designation of the HPAF lines in the two studies may overlap, as the literature use the two labels interchangeably (see for example, the cell line synonyms here, https://www.cytion.com/us/HPAF-II-Cells/305088 (accessed on 2 February 2025)). However, the HPAF-II designation does refer to a line of unlimited proliferative capacity, which was derived from HPAF-1 cells in static culture (http://en.ldraft.com/qdmore/189.html (accessed on 7 February 2025)). While we did not examine orthotopic models (but plan to use them in future studies), F5-FNIR distributed similarly to the cell-derived flank tumors, while showing no accumulation in PANC-1 MUC4^−^ tumors. After the initial studies proving tumor-selective deposition of Capan-1 and HPAF-II, we subsequently found a third cell line, BxPC3, which also displayed significant enhanced tumor accumulation of 5-FNIR relative to IgG isotype control. An important aspect of the work is the homogeneity of the antibody–dye conjugates. Contamination with higher-order structures, residual free dyes, or those of varying degrees of DOL could affect non-specific off-target binding.

The ultimate goal of our work is to develop these tools into therapeutic modalities that may ease the suffering of patients with PDAC tumors. We have shown that F5 alone does not possess any antibody-based cytotoxicity, i.e., Antibody-Dependent Cellular Cytotoxicity (ADCC) or Complement-Derived Cytotoxicity (CDC). While there are ways to modify mAbs to become more toxic to target cells through engineering the Fc portion of the protein, having highly selective antibodies in pancreatic tumor cells may lead to many other forms of targeted therapy such as ADCs, CAR-T cells, and BITEs. Now that we have shown that F5 can target PDACs in vivo, we are exploring several of these modalities to hopefully produce new drug agents with improved efficacy over current therapies. This is being pursued through rational selection of new neoantigen glycopeptides and the production of even more useful mAbs in the future. We are actively pursuing these directions and will report our results in due course.

## 4. Materials and Methods

Purchased Materials. Pierce^®^ Maleimide-Activated 96-Well Plates were purchased from Thermo scientific (Frederick, MD, USA. Solutions and buffers were from Sigma Aldrich (St Louis, MS, USA) or Thermo Scientific unless otherwise specified. Frederick Near-Infrared (FNIR) fluorescent dye was a gift from Dr. Martin Schnermann, CBL.

Cell Lines. The human pancreatic cancer cell lines HPAF-II, BxPC-3, Capan-1, and PANC-1 were obtained from the American Type Culture Collection (ATCC, Manassas, VA, USA). Capan-1 was maintained in IMDM media supplemented with 20%FBS, HPAF-2 in EMEM media with 10% FBS, while BxPC-3 and PANC-1 were maintained in DMEM media supplemented with 10% FBS. All the cells were grown at 37 °C with 5% CO_2_.

Synthesis of F5-FNIR conjugate. The F5 mAb (200 μL of a 5 mg/mL stock solution in PBS) was added to 100 μL of 50 mM PBS (pH 8.5) in a 1.5 mL microcentrifuge tube. In a second 1.5 mL tube, a 10 mM stock solution of FNIR-NHS Tag in DMSO (2, 4, and 7.5 eq. for Degree of Labeling (DOL) 1, 2, and 4, respectively) was quickly premixed with 100 μL PBS (pH 8.5) and immediately transferred to the F5 solution. The resulting mixture was incubated at room temperature for 1 h in a shaker. The solution was eluted through a PBS (pH 7.4)-equilibrated Zeba spin DS column (7K MWCO, Thermo Fisher Scientific) to remove unreacted free dye. A second overnight dialysis was performed to remove nonspecifically bound dye from the antibody. Finally, the solution was concentrated using Amicon ultra centrifugal filter units. (3K MWCO, Millipore/Sigma, St Louis, MS, USA). Preparation of FNIR-labeled isotype control antibody was performed identically to the above procedure. For the later in vivo studies for PDX models, these antibody–dye conjugates were subjected to the additional purification by SEC using an AKTA-go FPLC system (Cytiva Life Sciences, Marlborough, MA, USA) to remove any aggregates or residual dyes.

Characterization of antibody–dye conjugate. UV–visible spectroscopy was used to confirm the conjugation of FNIR to the antibody and to calculate degree of labeling (DOL). Absorbance was obtained using an NP80 Nanophotometer (Implen, Munich, Germany). DOL was determined by the equation: DOL = (A_765_/ε_dye_)/[(A_280_ − 0.05 × A_765_)]/ε_protein_(1)
where A_280_ and A_765_ are the absorptions at 280 nm and 765 nm, respectively, and ε is the molar absorption coefficient. The protein concentration (mg/mL) was calculated by the equation:[(A_280_ − 0.05 × A_765_)/ε_protein_] × MW_protein_ × dilution factor(2)

SDS page was performed to confirm the presence of covalent binding of dye to the antibody. Approximately 5 µg of conjugated and native antibodies was loaded on 4–12% gel under non-reductive conditions in running buffer at 200 V for 45 min with 1:4 (*v*/*v*) solution of LDS sample buffer and 50 mM PBS, pH 7.4. Fluorescence images were obtained using ImageQuant LAS 4000 (GE) (Chicago, IL, USA) with Cy5 excitation. The gel was subsequently Coomassie stained and imaged.

### 4.1. Binding Study of F5-FNIR Conjugate

*Surface Plasmon Resonance* (*SPR*). Binding kinetics was measured by Surface Plasmon Resonance (OpenSPR, Nicoyalife, Waterloo, ON, Canada). 5TF MUC4 peptide with a thiol linker was immobilized on the gold sensor chip (Nicoyalife). After immobilization, various concentrations (10–100 nM) of F5 and F5-FNIR prepared in PBS buffer containing 0.005% Tween 20 was slowly flowed over the sensor chip to allow interaction. After subtracting the background, the signal response vs. time curve was obtained; binding kinetic parameters were obtained by fitting the curve to a one-to-one binding model using TraceDrawer software (https://tracedrawer.com/ (URL accessed on 23 March 2025)).

*ELISA Assays*. ELISA was performed using the Pierce^®^ Maleimide-Activated 96-Well Plates (Thermo scientific). MUC4 glycopeptide with thiol linker was incubated with TCEP resin to obtain maximum free thiols. The MUC4 glycopeptide with linker was prepared in Binding Buffer (0.1 M sodium phosphate, 0.15 M sodium chloride, 10 mM EDTA; pH 7.2) at concentrations of 1–50 µg/mL. These glycopeptide solutions (100 µL) were added to each well and incubated overnight at 4 °C followed by washing three times 200 µL of Wash Buffer (0.1 M sodium phosphate, 0.15 M sodium chloride, 0.05% Tween^®^-20 Detergent; pH 7.2) for each wash. Then 200 µL of cysteine solution (10 µg/mL) was added to each well and incubated for 1 h to inactivate excess maleimide groups. Wells were washed and 100 µL of F5 or F5-FNIR of various concentrations were added in each well and incubated for 1 h at room temperature. The wells were washed and 100 µL of the secondary antibody conjugated to AP was added to each well and incubated for 1 h. After incubation, the wells were washed, and the enzyme substrate (PNPP) was added, with the absorbance measured at 405 nm using a plate reader. Absorbance vs. concentration was plotted using the Prism v. 10 (GraphPad, Boston, MA, USA).

*Western Blot.* Cell lysates were collected using cell lysis buffer (NP-40 lysis buffer) containing protease and phosphatase inhibitors. The protein concentration in each sample was estimated by UV spectroscopy on a Thermo NanoDrop spectrophotometer (Thermo Fisher Scientific, Waltham, MA, USA). About 40 μg of protein from each sample was analyzed by SDS-PAGE (2 h elution at 90 V). The eluted proteins were transferred from the gel to a polyvinylidene difluoride (PVDF) membrane and blocked with 4% BSA for 2 h. F5, as a primary monoclonal antibody (1:1000 dilution), was incubated on the membrane at 37 °C for 2 h and washed several times. Horseradish peroxidase-conjugated secondary antibody was incubated with the membrane for 1 h, washed, and detected by SuperSignal™ West Pico PLUS Chemiluminescent Substrate (Thermo Fisher Scientific (Frederick, MD, USA). Western blots were imaged using an ImageQuant LAS 4000.

Biodistribution Studies in Tumor CDX and PDX Models. In vivo studies were performed according to Animal Care and Use committee guidelines at Frederick National Laboratory for Cancer Research (Frederick, MD, USA). Frederick National Laboratory for Cancer Research is accredited by the American Association for Accreditation of Laboratory Animal Care (AAALAC) International and follows the Public Health Service Policy for the Care and Use of Laboratory Animals. Animal care was provided in accordance with the procedures outlined in the “Guide for Care and Use of Laboratory Animals” (National Research Council; 2011; National Academies Press; Washington, DC, USA). Approximately 4–7-week-old female athymic nude mice were purchased from Charles River Laboratories International, Inc. (Frederick, MD, USA). Capan-1, HPAF-II, BxPC-3, and PANC-1 cells were used to develop the cell-derived xenograft model.

*Cell-Derived Xenograft* (*CDX*) *Tumor Models.* Two million cells from both MUC4^+^ cell lines (Capan-1, HPAF-II, and BxPC3) and the single MUC4^−^ cell line (PANC-1) were injected subcutaneously into the flanks of athymic nude mice. Tumor sizes were assessed twice weekly and after 3 weeks post-cell injections, animals were divided into groups of PBS-only and F5-FNIR-treated. For the Capan-1 xenograft model, one additional isotype control group (IgG-FNIR) was also included. Additional biodistribution studies were performed similarly using the BxPC-3 cell line for further validation of F5-FNIR tumor localization in comparison with IgG-FNIR.

*Patient-Derived Xenograft* (*PDX*) *Tumor Models.* Two PDX tissues with differential MUC4 expression were obtained from NCI Patient-Derived Models Repository (PDMR). The two PDX tissues (133R and 295R) were confirmed for differential MUC4 expressions by immunohistochemistry (IHC) staining using F5 as the primary antibody (1:1000 dilution) and compared with isotype control. F5 showed very strong staining for high MUC4-expressing PDX (133R) while showing minimal staining on low MUC4-expressing 295R. Eight-to-twelve-week-old NOD *scid* gamma mice (NSG^TM^, stock number #005557) were purchased from The Jackson Laboratory (Bar Harbor, ME, USA). These animals were then implanted with PDX tissues subcutaneously into the left flank with a single ~1 mm × 1 mm × 1 mm fragment of the corresponding PDX tissue. Mice were monitored weekly by palpation, tumors were measured using standard calipers, and tumor volumes were computed using the equation Volume = ½ × Length × Width^2^. Mice carrying tumors that reached the volume of 250–300 mm^3^ were randomized into three treatment arms and injected with F5 vs. control antibodies, as indicated above for the CDX models. In this study, the 295R PDX tissue growth was very slow and could not produce tumors that would be viable for the biodistribution, thus only the MUC4^+^ 133R PDX model was used. For animal administration, PBS (100 µL), F5-FNIR (50 µg in 100 µL), or IgG-FNIR (50 µg in 100 µL) were injected intravenously, via tail vein, in three different groups of 133R PDX model-carrying mice. Animals were imaged at different time intervals (4 h, 24 h, 48 h, and 72 h) upon administration of F5-FNIR antibody or control reagents. Fluorescence was monitored by employing the IVIS spectrum imager (PerkinElmer Inc., Waltham, MA, USA). Images were acquired and processed using Living Image software (v. 4.8.2). At the end of study, animals were sacrificed and different organs including tumors were isolated and fluorescence was monitored using IVIS spectrum imager.

### 4.2. Statistical Analysis

We have employed two-way Anova for the data provided in the paper to calculate the statistical significance between groups (* denotes *p*-values < 0.05, denotes *p*-values < 0.01 **, *** denotes *p*-values < 0.001). The statistical analysis was performed using GraphPad Prism 10 software.

## Data Availability

The original contributions presented in this study are included in the article/Appendix A. Further inquiries can be directed to the corresponding author(s). The raw data supporting the conclusions of this article will be made available by the authors upon request.

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
