# Peer review of "Biodistribution of a Mucin 4-Selective Monoclonal Antibody: Defining a Potential Therapeutic Agent Against Pancreatic Cancer"

_ijms, 2025, doi:10.3390/ijms26136042_

Round 1

Reviewer 1 Report

Comments and Suggestions for Authors

Found in attachment.

Author Response

We thank the reviewer's for their useful comments to improve our manuscript. We will answer all comments below in turn:

Reviewer #1:

  • F5 cannot induce ADCC or CDC. A rationale is required?

Assays we performed showed that F5 had negligible cytotoxicity towards the tumor cells mediated via ADCC or CDC. The most likely reasons for this are: 1) Insufficient interaction with Fc receptors on relevant immune cells, possibly caused by deficient or incompatible Fc glycosylation at asparagine (Asn) 297, and 2) the F5 mAb is a mouse IgG1 which has been reported to have less CDC or ADCC toxicity compared with isotypes. Re-engineering the F5 antibody in the context of a different isotype, such as IgG2a with modifications to the N297 glycosylation, or employing a known ‘LALAPG’ modification of the antibody’s heavy chain that lacks the Fc receptor binding affinity may further mitigate unspecific cytotoxicity. We are plan to explore these options in future preclinical validation of these reagents in vivo.

  • Are there data on which region of F5 is glycosylated? NHS conjugation is not specific, so there can be conjugation to the variable region explaining the mild decrease in binding. Is the F5-dye construct homogeneous?

We thank the reviewer for bringing this point up; as alluded to in point #1 above, we have not determined the exact glycosylation pattern at Asn297. It is also plausible that this glycosylation may vary depending on the tissue culture conditions and a choice of a cell line employed for recombinant production of F5 protein. As pointed out by the reviewer, the NHS conjugation is non-specific and may affect the binding of antibody to its antigen if the conjugation occurs on variable regions involved in epitope binding. In our case, however, we believe that conjugation is not impacting the binding region of antibody since antibody-dye conjugates have very comparable binding affinity vs.  the non-labelled F5 antibody towards the antigen. If the conjugation site had been located on the variable regions we would expect a much more significant decrease in affinity. The F5-FNIR is obtained via amine coupling to NHS-modified dye molecules, and hence conjugation will produce heterogenous mixtures. Thus, the degree of labelling obtained may be the average of the different number of dye moieties connected to each antibody molecule. However, the final constructs used for experimentation are “homogeneous” in the sense that we perform extensive purification and removal of excess dye before any in vitro or in vivo work. In future work, however, it is our intent to explore selective conjugation to specific F5 mAb residues.

  • The way the loading was calculated (by fluorescence intensity) seems highly sensitive to contamination with trace amounts of the dye. Is there a reason for not using direct measurements, like MALDI? As is, the procedure might be harder to replicate.

The loading was calculated based on UV absorbance of the dye and antibody. Since Amax of the dye is close to 765 nm, whereas the F5 is detectable only at the standard aromatic protein absorbance at 280 nm, interference from the dye is minimal and the method we employed is widely used for calculating the DOL of antibody-dye conjugates. As the reviewer suggested, this method may be sensitive towards free dye contamination. Our extensive multi-step purifications protocols for labelled F5, including size-specific ultrafiltration, dialysis, and size exclusion chromatography should reliably eliminate any contamination by free dye material. Hence, we believe our DOL calculations to be accurate and thus did not perform MALDI. At these protein sizes, MALDI also provides what could be considered a range of molecular weighs and hence is comparable to our absorbance method. To answer the reviewer more completely, we have performed MALDI on our conjugates and compared the calculations to the absorbance method. The results matched well, and we have included them in the text and added a supporting figure which illustrates and compares these data.

  • In Fig. 1, the mucin only has the Tf antigen and is highly glycosylated. While this reviewer does not know MUC4 specifically, MUC1 tends to be underglycosylated and has a larger diversity of glycans: Tf, sTf, Tn, sTn, Lea, Lex. etc. If this is the same for MUC4, the figure is then misleading.

The reviewer is correct here and many TACAs are indeed found on the MUC4 glycoprotein. The figure was simply a schematic illustration of a generic cell-surface mucin. We have adjusted the figure and added a more descriptive legend to more accurately describe what we were attempting to convey.

  • F5 is never clearly described as an IgG. While this is obvious from the isotype controls, early clarification would improve readability.

We have added this description in the Introduction—thanks!

  • In Figure 4, the tumors have been circled. The circles need to be more visible. A quick annotation of the organs to show tumors, liver etc. would nicely complement the following discussion on biodistribution. Higher resolution pictures, if the current one is not an instrument limitation, would also be appropriate.

We have made improvements to the Figure as aptly suggested by the reviewer.

Reviewer 2 Report

Comments and Suggestions for Authors

The topic of the article is relatively innovative, but the workload is not substantial, and the biggest issue is the obvious problems with data analysis, which are very rough. For example, in the WB image of Figure 2, there is a clear inversion. In Figure 4, the author intends to convey that the graph on the right shows raw data of total radiance efficiency, depicting a much greater distribution to MUC4+ CDX tumors compared to MUC4- tumors. However, the statistical difference is marked based on time differences instead. Additionally, in Figure 6, the immunohistochemistry image only includes a zoomed-in version, with no original image or both original and zoomed versions, and no statistical analysis is provided. 

Author Response

Reviewer 2 comments:

General: We respectfully disagree with many of the comments and the idea that nearly everything about this manuscript  “Must be Improved”.  First, the senior author (JJB) is a native English speaker and has written hundreds of articles. We feel there is very little, if anything wrong with the English and it does not need improvement. If there are specific instances where a word or sentence could be stated more clearly, we would happily change them, but this reviewer did not provide any examples so we will not respond further. In addition, since the reviewer did not provide specific examples of what needed revising regarding the Introduction, references, adequate method descriptions, clearly stating results and how the results support the conclusions, we will not further address any of these issues. The following is our response to the specific queries of the review:  

The topic of the article is relatively innovative, but the workload is not substantial, and the biggest issue is the obvious problems with data analysis, which are very rough. For example, in the WB image of Figure 2, there is a clear inversion. In Figure 4, the author intends to convey that the graph on the right shows raw data of total radiance efficiency, depicting a much greater distribution to MUC4+ CDX tumors compared to MUC4- tumors. However, the statistical difference is marked based on time differences instead. Additionally, in Figure 6, the immunohistochemistry image only includes a zoomed-in version, with no original image or both original and zoomed versions, and no statistical analysis is provided. 

We thank the reviewer for these comments. However, there is still a severe lack of examples why the workload is not “substantial” and the “obvious problem with data analysis” except for the three examples of the figures the reviewer refers to. We will take each one in turn:

1) The western blot in Figure 2 will be revised for clarity. It is not inverted. The bands on the bottom are simply fluorescent components in the LDS sample buffer that are detected in the analysis. We apologize that the molecular weight ladder on the left does not have MW markers which we should have included…these are now in the new figure. The Coommasie stained gel on the left was simply to show there was protein in each sample since only the dye-labeled molecules would show up in the fluorescent-detected gel. The legend has been updated to clarify these points.

2) The statistics in Figure 4 were performed for EACH time period, so for each point on the graphs and it showed statistical significance between the points on the upper curve (MUC4+ radiance) relative to the lower (curve MUC4- radiance). This statistical analysis was not based on time, but on the difference between MUC4+ ad MUC4- at each different time point. We have added text to try and describe this more clearly. 

3) We have modified Figure 6 to include both whole tissue (Zoomed out) and a zoomed in view. We have re-stained the tissue and have calculated statistics. A more complete description of these changes is now included in the text as well as the statistical analysis of the additional newly stained tissue. 

Reviewer 3 Report

Comments and Suggestions for Authors

Dahal et al. have developed and standardized the biodistribution of a Mucin 4-selective monoclonal antibody in a pancreatic cancer model. This is a strong proof-of-concept study; however, several concerns need to be addressed:

  1. The authors claim that the F5-FNIR conjugate binds selectively to Mucin 4. To support this, authors should provide data demonstrating both the specificity and selectivity of F5-FNIR toward Mucin 4. This could be achieved through fluorescent co-localization studies or protein-binding assays.
  2. The authors use HPAF-II and Capan1 cell lines (MUC4-positive) and Panc1 (MUC4-negative) as controls. To strengthen their claims, they should consider performing MUC4 knockout in HPAF-II and Capan1 and demonstrate that F5-FNIR binding is reduced or abolished, thereby confirming the antibody's selectivity.
  3. In Figure 5B, both the liver and tumor show ex vivo fluorescence signals. The authors should elaborate on why the liver exhibits higher dye accumulation compared to tumors—this may reflect dye metabolism, clearance, or off-target retention and warrants further explanation.
  4. In Figure 6, while MUC4 expression is expected in pancreatic cancer, PDX tumors in panels C–D show no MUC4 staining. The authors should clarify this discrepancy and provide possible explanations for the absence of signal.
  5. Lastly, the number of animals (N) used in each experiment should be clearly indicated in the figure legends to ensure transparency and reproducibility.

Round 2

Reviewer 2 Report

Comments and Suggestions for Authors

The figures and data in this article still have significant issues. For example, how is the statistical analysis in Figure 7 valid?

Author Response

We certainly appreciate the reviewer's comments but we respectfully take issues with the following:

1) The fact that this reviewer feels the English needs improving. We believe this to be patently untrue and we would be happy to have another native English speaker attest to this fact.

2) The fact that the reviewer thinks "everything" in the paper MUST be improved. With all due respect, we feel that (a) The references are relevant and complete, (b) the research design is appropriate and performed with scientific rigor, (c) The methods are adequately described as can readily be seen in the Materials and Methods and any omission or lack of clarity in this section has been addressed in the revision (d) The results are clearly presented, except for perhaps some very minor clarity issues and statistical analysis which were corrected and/or included in the revision and (e) The conclusions are highly supported by the results. We are not concluding anything frivolous or speculative--ONLY what we observed.

The reviewer is welcome to list IN DETAIL all that is "wrong" with the paper and we will be happy to respond in kind. 

Reviewer #2:

The figures and data in this article still have significant issues. For example, how is the statistical analysis in Figure 7 valid?

We would like to thank the reviewer for the comments. We address the comments by providing a comparison of the statistics obtained by two-way Anova, one way-Anova and T-test for Figure 7. We have employed two-way Anova for the data provided in the paper as we wanted to see the statistical significance between groups over different time points. 

Different statistical analysis revealed that the overall conclusion of our data remains unchanged although some statistical power changes for some time points. We have provided an original analysis of Figure 7 , and also for 4 and compared them with standard one-way Anova and T-test.  One-way Anova decreased the P-value for most of the time points compared to two-way Anova making it more statistically significant. We have included and additional file with our submission that show all the raw statistical data that was calculated( in the attachment ). We also have included a separate Materials And Methods statement about statistics, and have edited the legend to Figure 7 to include what we observed in our analysis 

We hope now our paper is acceptable for publication. If there are any other details we need to attend to, please let us know!

Reviewer 3 Report

Comments and Suggestions for Authors

All comments have been addressed.

Comments on the Quality of English Language

Good